# Trans-Fatty Acids in Fast-Food and Intake Assessment for Yerevan’s Population, Armenia

**DOI:** 10.3390/foods11091294

**Published:** 2022-04-29

**Authors:** Davit Pipoyan, Meline Beglaryan, Seda Stepanyan, Nicolò Merendino

**Affiliations:** 1Center for Ecological-Noosphere Studies of NAS RA, Abovyan 68, Yerevan 0025, Armenia; david.pipoyan@cens.am (D.P.); meline.beglaryan@cens.am (M.B.); seda.stepanyan@cens.am (S.S.); 2Department of Ecological and Biological Sciences (DEB), Tuscia University, Largo dell’Università snc, 01100 Viterbo, Italy

**Keywords:** daily intake, fast-food, gas chromatography-mass spectrometry, risk, trans-fatty acid

## Abstract

There are stringent regulations applicable for trans-fatty acid (TFA) limitations from food supply across the world. However, in Armenia, there is a scarcity of data on TFA content in food products and their consumption levels. Considering that fast-food is among the major contributors to TFA intake, this study aims to assess the dietary exposure of TFAs through the consumption of fast-food in Yerevan, Armenia. Eleven types of fast-food were included in the study. The Food Frequency Questionnaire (FFQ) was used to evaluate daily fast-food consumption. TFA contents in samples were determined using gas chromatography-mass spectrometry. Mean daily fast-food consumption values ranged from 14.68 g/day to 76.09 g/day, with popcorn as the lowest and pastry as the highest consumed food. The study results indicate that the aggregate average daily intake (DI) of TFA is 0.303 g/day. Even though TFA DI values do not exceed the WHO limit of 1%, they substantially contribute to daily TFA intake and may exceed the limit when combined with other foods. Hence, it is recommended to carry out continuous monitoring of TFA content in the food supply to ensure consumer health protection.

## 1. Introduction

The increased consumption of fast-food has become a significant public concern worldwide since fast-foods are typically rich in trans-fatty acids (TFAs) which are associated with various health diseases [1,2]. Trans-fatty acids are unsaturated fatty acids containing at least one double bond in their trans configuration. TFAs may occur naturally or as a result of various chemical processes. TFAs in diet may derive from the main two sources: naturally occurring TFAs and industrially produced TFAs. The naturally occurring TFAs are produced by the bacterial transformation of unsaturated fatty acids in the rumen of animals like cattle and goats. The industrial TFAs are produced by the industrial hydrogenation, deodorization of unsaturated vegetable/fish oils, and heating and frying of oils at high temperatures. The industrial TFAs contain trans isomers of oleic acid (C18:1 cis-9), the major one being elaidic acid (C18:1 trans-9). The ruminant TFA in milk and meat consists mainly of vaccenic acid (C18:1 trans-11) and conjugated linoleic acid (CLA or 9-cis, 11-trans-C18:2). Based on some risk markers, the ruminant TFAs are less harmful than the industrial TFAs (e.g., for HDL) or present a neutral effect (e.g., for TG, glycemia, insulin resistance, blood pressure) [3,4].

Most commercial foods contain TFAs, produced as a result of industrial hydrogenation. This process improves food’s texture, plasticity, and increases its stability for long shelf life [5]. Many comprehensive studies confirm the presence of trans-fatty acids in industrially produced foods, particularly in various fast-food items [6,7].

In recent decades, fast-food consumption has increased worldwide [8,9]. As a consequence of the increased intake of fast-food, the dietary intake of TFAs has been increasing in many countries [10,11]. A high TFA intake raises significant health concerns and contributes to increased morbidity and mortality. There is consistent evidence of industrial trans-fatty acids’ adverse health effects, particularly on the levels of low-density lipoprotein cholesterol in the blood, coronary heart disease, cancer, infertility, obesity, Alzheimer’s disease, allergy, and type 2 diabetes [12,13,14,15,16]. Moreover, the consumption of industrial TFAs increases the risk of atherosclerosis, apoptosis, and inflammation [17]. According to Stender and Dyerberg, there is a positive relationship between trans-fatty acid intake and the incidence of breast and large intestine cancer [18]. High intakes of TFAs have been reported to be an independent risk factor for cardiovascular diseases [19,20]. According to the European Cardiovascular Disease (CVD) Statistics, trans-fat consumption significantly increases the CVD risk, which is the main cause of death for Europeans under the age of 65 [21]. World Health Organization (WHO) reports that a 2% increase in energy from trans-fats leads to a 25% increase in the risk of death from CVD. It is noteworthy that according to the WHO report, every year more than half a million deaths of people from cardiovascular disease can be attributed to the intake of TFAs [16,22].

To control TFA levels in food and to protect health and save lives, throughout the years, various approaches have been implemented to reduce TFA amounts in foods. Limitations on the content of industrialized TFA have been made in several countries, such as Denmark, Austria, Switzerland, Iceland, Norway, Hungary, Sweden, Latvia, and Georgia. Others (e.g., USA, Brazil) imposed mandatory labeling. Year by year the latest national and international recommendations suggest that dietary intakes of TFA should be as low as possible [3]. According to European Food Safety Authority’s (EFSA) opinion, in European Union, foods intended for consumer use shall have to contain less than 2 g of industrial TFAs per 100 g of fat. Commission Regulation (EU) 2019/649 states that the trans-fat other than trans-fat naturally occurring in the fat of animal origin must be a maximum 2 g per 100 g of fat in food intended for the final consumer and food intended for supply to retail [23]. In several countries, such as UK and France, the TFA limit is also set at 2% of overall energy [3,24]. In May 2018, WHO introduced an action package to support governments to eliminate industrially produced TFA from the global food supply by 2023. The action package recommends that the average intake of trans-fats should be less than 1% of total energy. Moreover, WHO called for a replacement of TFA with healthier oils and fats, to be achieved through policy and regulation, while establishing monitoring systems and creating awareness among policymakers, industry, and the public [22].

Regulatory requirements that limit TFA in oils and fats came into effect in Eurasian Economic Union (EAEU) member countries, including Armenia [25]. Compared to the EU, the legal limits on trans-fat in EAEU seem to be less stringent. The EAEU was the first trade bloc globally to restrict TFA, with a 20% TFA limit in specific fat products that came into effect in January 2015. However, from January 2018, according to the recent changes in the Technical Regulation of the Customs Union (TR CU 024/2011) [25], the limit for “trans-isomers of fatty acids” changed from 20% to 2% of the product’s total fat content. Currently, it is mandatory to indicate the amount of the product’s fat content on the package. It is important to note that there are no labeling requirements for other products rather than fats and oils in EAEU, which is a fundamental difference compared to the EU regulations. Moreover, studies still need to be carried out to show if TFA needs to be regulated in other food products as well that use fat and oil as their raw materials.

Like most countries, Armenia is severely affected by high non-communicable diseases (NCDs) prevalence. Over 90% of all deaths are attributable to NCDs [26]. Fast-food is a major driver of high NCDs, and in recent years, the number of fast-food restaurants and selling points increased in the country. Hence, it is particularly important to study and assess the health risks of TFAs among Armenians and to carry out appropriate preventive measures. In this study fast-food is defined as “processed food that can be prepared easily and is served as a quick meal on the premises or is taken away”. It is noteworthy that there are no reports or studies about fast-food consumption patterns in Armenia. Moreover, risk assessment of trans-fatty acids through consumption of fast-food has never been carried out in Armenia before. Hence, for the first time, the present study aims to investigate fast-food consumption patterns in Yerevan, Armenia, and assess the dietary exposure of trans-fatty acids.

## 2. Materials and Methods

### 2.1. Food Sampling

Fast-food sampling was done in 2020 following WHO’s protocol [27]. All the fast-food samples were randomly collected from 4 major grocery stores and street vendors in Yerevan. The number of the collected individual samples (8 sub-samples per product) covered the main markets in Yerevan. Before analysis, the individual samples were pooled to create representative composite samples for each type of fast-food. The composite samples were ground and homogenized thoroughly. Overall, eleven types of fast-food were included in the study: fried chicken, shawarma, pizza, hot-dog and burger, pie, khachapuri, fries, chips, popcorn, pastry, and pancake/doughnut.

### 2.2. Analysis of TFAs

The ISO 12966-4:2015 standard was used for the determination of trans-fatty acids in the selected food items (g/100 g of product) using the gas chromatography (GC) method [28]. This method is developed for the determination of fatty acid methyl esters (FAMEs) derived by transesterification or esterification from fats, oils, and fatty acids by capillary gas chromatography. The tests were performed by a laboratory of Standard Dialog LLC which has an international accreditation according to ISO 17025. The preparation of the methyl esters of fatty acids was done using methods specified in ISO 12966-2:2017 standard [29]. The analysis of prepared methyl esters performed by GC and clear separation of 37 component mix and method requirements for selectivity of the column were reached. The “CP-Sil 88 for FAME 100 m × 0.25 mm × 0.2 µm 5 inch” column (model: CP7489I5, manufactured by: Agilent Technologies, Santa Clara, CA, USA) was used. For the quantification of the fatty acids, in grams per 100 g, the Supelco 37 Component FAME Mix certified reference material (TraceCERT^®^, Sigma-Aldrich Chemie GmbH, Darmstadt, Germany) was used. Shimadzu 2010 GC system injection volume was 1 μL.

Quality assurance standard procedures were carried out to ensure the reliability of the results. The determination of TFA content in samples was performed with three replications. The total content of TFAs was reported as a sum of the detected isomers.

### 2.3. Food Consumption Data Collection

The Food Frequency Questionnaire (FFQ) was used for investigating dietary consumption patterns of fast-food among the Yerevan population. The questionnaire was developed based on the data from the 24-h recall (24HR) which is described in detail in Pipoyan et al., 2020 [30]. This does not require ethical approval from an ethics committee, as it did not include the collection of identifiers (e.g., name, surname, etc.) from the interviewed population.

FFQ surveys were carried out in the fall of 2019. Data was collected verbally and in person by well-trained interviewers. In total, 423 respondents aged from 18 to 65 took part in the survey. This sample size exceeds the recommendation of 260 adults (18–64 years) made by EFSA [31].

FFQ fast-food list was developed based on the most consumed fast-foods cited in the last 24 hr. The final list included eleven types of food items. The questions in the survey included information about the consumption frequency, portion size, and food source. Data regarding gender (43.7% women and 56.3% men), education level (90% received higher education), and income level (66% refused to answer this question) were collected. During the process of conducting surveys, equal distribution of population among Yerevan city administrative districts and age groups was taken into consideration.

The consumption frequency response options for the FFQ food list were: “never”, “once per month”, “2–3 times per month”, “once per week”, “2–4 times per week”, “every day”, “other” (more/less frequent). According to the type of fast-food, several portion sizes were determined to record food consumption: grams, units, pieces, handfuls, servings, boxes, cups, bags, and candy bars. To aid the reporting process, detailed instructions were given by interviewers and when needed, pictorial depictions of serving measures (i.e., different sized boxes, bags, and cups) were presented to the respondents.

### 2.4. Data Analysis

Data were analyzed by SPSS software (SPSS Inc., version 22.0, New York, NY, USA). Descriptive statistics were calculated. Since fast-food consumption values were non-normally distributed, the K-means cluster analysis method was applied. This method was shown to be effective especially for the analysis of dietary patterns in a large population using FFQs [32]. The optimal number of clusters was determined experimentally, through 10 iterations. For each iteration, the sum of squared deviations was identified using the ANOVA table. To calculate the daily fast-food consumption, the daily frequency of consumption of each food was multiplied by the portion size of that food.

### 2.5. Daily Intake of TFA

Daily intake (*DI*) of TFA (g/day) was calculated through the following equation:(1)DI=C×IR,
where *C* is the mean content of TFA in all the studied fast-food products (g in 100 g), and *IR* is the daily consumption of fast-food (g/day).

The individual TFA intake was calculated as the percentage of total energy using the following formula [33]:(2)$$E\%=\frac{DI\times 9}{DE}\times 100$$
where *E*% is the individual TFA intake as the percentage of total energy, *DI* is the individual TFA intake per day (g/day), and *DE* is the individual total dietary energy intake (kcal). The energy transfer index of TFA was 9 kcal/g. The amount of dietary energy received from food consumption is approximately 2047 kcal/capita/day [34].

## 3. Results and Discussion

### 3.1. TFA Content in Fast-Food

The study results indicate that the total TFA contents in fast-food samples range from 0.002 to 0.3 g (g/100 g). Among the studied fast-foods, pizza, hot-dog and burger, fries, and pancake had the highest TFA content, while popcorn, pastry, and chips had the lowest TFA content (Figure 1). Currently, there is no regulation regarding TFA content in fast-food in Armenia.

The results of the current study are consistent with other investigations. A study conducted in 2017 reported data on Spanish fast-food (French fries), showing low TFA contents that ranged from 0.04 to 0.11 g TFA/100 g product. On the other hand, in this study, the TFA content detected in fries was 0.083 g/100 g. Stender reported the TFA content in prepackaged biscuits/cakes/wafers purchased from the three largest supermarkets in Armenia [35]. According to the results, there were 99 different packages of biscuits/cakes/wafers with more than 2% trans-fat per 100 g fat in Armenia. In the frame of the investigation on TFAs in Polish pastry, a big diversity (0.02 to 3.49 g per 100 g of product) of total TFA content was found [36]. The research results on bakery products (cookies, biscuits, wafers) from the Slovenian market showed high variability in TFA content with a mean of 1.91 g per 100 g of product [37]. A relatively small mean content (0.008 g per 100 g of product) of TFA was reported for confectionery and pastries studied in Spain [38]. In 2008, the Food Safety Authority of Ireland commissioned a study of the TFA content of fast-foods collected from 12 Irish restaurants. A high of 0.6 g TFA per 100 g of product was recorded in burger products. Lower levels ranging from 0.2–0.5 g TFA per 100 g of product were found in different fast-food products, including pizza, fries, and sandwiches [39]. Overall, the levels of TFA in fast-foods are higher than the ones reported in this study. According to a study conducted in Germany, 0.04 g TFA per 100 g of product was found in French fries [40]. The detected content is twice lower than the one reported in this study.

### 3.2. Fast-Food Consumption

The FFQ study sample was drawn from the adult population of Yerevan, whose average age was 31.91 ± 11.85 years. The average weight for males was 74.76 ± 12.45 kg and for females 58.23 ± 12.02 kg.

Based on survey analysis, all the study participants consumed at least one type of fast-food. Most of the food types are consumed mainly once a week. Most of the respondents do not use chips, popcorn, and pancake. The results indicate that on average, the daily consumption values of fried chicken, shawarma, pizza, hot-dog and burger, pie, khachapuri, fries, chips, popcorn, pastry, and pancake are 53.09 g, 49.26 g, 33.88 g, 49.97 g, 34.97 g, 32.17 g, 27.94 g, 36.24 g, 14.68 g, 76.09 g, and 41.36 g, respectively. This implies that the most widely consumed fast-food products include pastry, fried chicken, shawarma, hot-dog, and burger (Figure 2).

In the frame of the study, the results of K-means cluster analysis revealed two cluster groups of fast-food consumers categorized in the order of consumption amount, from the lowest to the highest. Through K-means cluster analysis, the heterogeneous consumption data was grouped into homogenous clusters for each fast-food. Figure 3a,b summarizes the average daily consumption amount and the percentage of consumers for two different cluster groups.

It should be noted that Cluster 1 is characterized by a low consumption value and it includes most of the consumers. Meanwhile, Cluster 2, having the highest consumption value includes only a small percentage of the population.

In the current study, the overall mean frequency of fast-food consumption is once (0.38–1.93) per week. A cross-sectional study conducted among adults in King County, US, reported a mean fast-food consumption of 2.07 times per week [41], which is more than the one reported in this study. Another study conducted among Iranian adults reported a mean fast-food consumption of 1.673 times per week, which is again higher compared with the result of the current investigation [42].

### 3.3. Daily Intake of TFA

To conduct a dietary exposure assessment, the daily intake of TFA was calculated for each fast-food. Table 1 summarizes daily intake values calculated for both the total average fast-food consumption and the two clusters of fast-food consumption.

Among all the studied food types, pizza, hot-dog and burger, and pancake have the highest contribution to daily TFA intake. Meanwhile, popcorn, chips, and pastry have the lowest contribution to daily TFA intake. Compared with WHO’s recommended level of less than 2.2 g/day for total TFA intake, none of the DI values exceed the limit [21].

DI values of TFA have been represented as a percentage of total energy intake. TFA intake through fried chicken, shawarma, pizza, hot-dog and burger, pie, khachapuri, fries, chips, popcorn, pastry, and pancake consumption accounts for 0.006%, 0.007%, 0.045%, 0.040%, 0.003%, 0.004%, 0.010%, 0.002%, 0.0001%, and 0.003% of the total energy intake, respectively (Table 1). Compared with WHO’s recommended level of 1% of total energy intake, these values do not exceed the limit and are not indicative of a major health concern [21].

Although the separate consumption of the studied food items does not warrant any concerns, their cumulative intake may be somewhat worrisome. Based on a statistical analysis of FFQ data, it was found that 7% of the respondents consume all eleven types of fast-food regularly. In the case of cumulative intake, the average DI of TFA is equal to 0.303 g/day, which is approximately 7 times lower than the WHO recommended level of 2.2 g/day. A similar pattern is present when representing TFA intake as a percentage of total energy. In the case of average fast-food consumption, TFA intake accounts for 0.134% of total energy intake, which is far below the level (less than 1% of total energy intake) set by WHO.

## 4. Conclusions

This was the first study towards the assessment of the total TFA daily intake through fast-food consumption among Yerevan’s adult population. In general, low levels of TFA were recorded in the investigated fast-food items, ranging from 0.002 to 0.3 g of TFA per 100 g of the product. Mean daily fast-food consumption values ranged from 14.68 g/day to 76.09 g/day, with popcorn as the lowest and pastry as the highest consumed food. The study results highlighted that the aggregate average daily intake (DI) of TFA (0.303 g/day) did not exceed the WHO limit of 1%. However, it was found that the consumption of fast-food alone can already contribute to the daily TFA intake of consumers.

The key findings of this study can serve as a basis for developing programs to reduce trans-fats to less than 1% of total energy intake and replace trans-fats with polyunsaturated fats. Considering the adverse health effects related to fast-food consumption, dietary advice and policy actions could be aimed at decreasing the consumption of these foods. Besides that, educating the population about limiting the consumption of foods that contain industrially produced trans-fats would be a crucial step to promoting a healthier diet.

National policies should be directed towards reducing incentives for the food industry to continue the production of TFA-rich foods. Taking into consideration the fact that TFA can also be present in other food types, it is crucial to have proper and continuous monitoring of TFA contents. Moreover, studies shall be conducted to investigate the adverse health effects of the aggregate TFA intake through the consumption of various food products.

## Figures and Tables

**Figure 1 foods-11-01294-f001:**
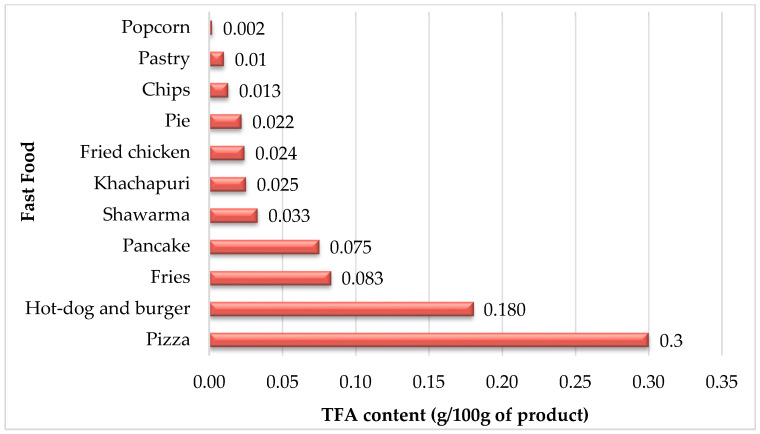
TFA Contents in Fast-Food (g/100 g).

**Figure 2 foods-11-01294-f002:**
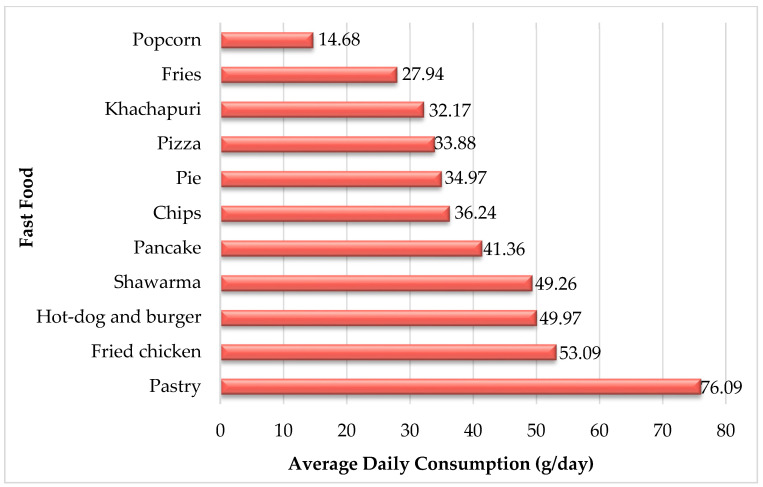
Average Daily Consumption of Fast-Food (g/day).

**Figure 3 foods-11-01294-f003:**
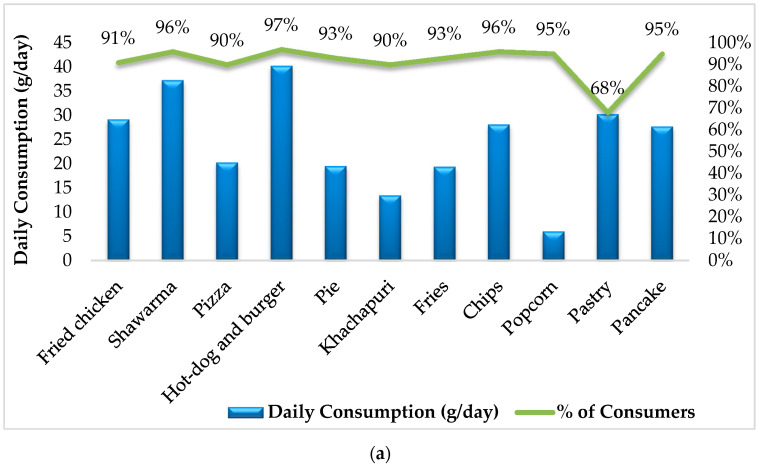
Daily Consumption (g/day) of Fast-Food for Cluster 1 (**a**) and Cluster 2 (**b**).

**Table 1 foods-11-01294-t001:** DI of TFA (g/day), TFA intake as a % of total Energy (E%).

Fast-Food	DI (g/Day)			E%		
	Average	Cluster		Average	Cluster	
		N1	N2		N1	N2
Fried chicken	0.013	0.007	0.053	0.006	0.003	0.023
Shawarma	0.016	0.012	0.116	0.007	0.005	0.051
Pizza	0.102	0.061	0.359	0.045	0.027	0.158
Hot-dog and burger	0.090	0.073	0.699	0.040	0.032	0.307
Pie	0.008	0.004	0.043	0.003	0.002	0.019
Khachapuri	0.008	0.003	0.023	0.004	0.001	0.010
Fries	0.023	0.016	0.101	0.010	0.007	0.044
Chips	0.005	0.004	0.032	0.002	0.002	0.014
Popcorn	0.000	0.000	0.001	0.000	0.000	0.001
Pastry	0.008	0.003	0.015	0.003	0.001	0.007
Pancake	0.031	0.021	0.175	0.014	0.009	0.077

## Data Availability

Data is contained within the article.

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
