# Peer review of "Trans-Fatty Acids in Fast-Food and Intake Assessment for Yerevan’s Population, Armenia"

_foods, 2022, doi:10.3390/foods11091294_

Round 1

Reviewer 1 Report

Trans-fatty acids in fast-food and intake assessment for Yerevan’s population, Armenia

Topic of manuscript is valuable, up- to date  and probably would gather great audience. However I have doubts regarding preparation of manuscript and description of results. I found some shortcomings, weaknesses and doubts which should be revised or/and clarified during the review. My general and specific comments are given below:

General comment:

Firstly, it is serious mistake to present information about trans fatty acids arising in biohydrogenation PUFA in rumen (in ruminants) and trans fatty acids (FAs) formed during industrial hydrogenation of vegetable fats (or heating during frying process) in one sentence (Lines 31-34). Please revise and include correct information from literature. Naturally arising trans FAs (from milk fat or from beef /lamb meat) not cause health problems. For example vaccenic acid (C18:1 11trans) has beneficiary effect. The serious cardiovascular problems resulted from consumption of products containing industrially hydrogenated fats.  You can find more information in other study.

Detailed comments:

Introduction:

  1. Lines: 35-38: Please include also the latest research findings about the presence and content of trans FAs and their functional role in many products: https://doi.org/10.3390/nu13093087
  2. Line 66 it should be also included information about Commission Regulation (EU) 2019/649 of 24 April 2019 amending Annex III to Regulation (EC) No 1925/2006 of the European Parliament and of the Council as regards trans fat, other than trans fat naturally occurring in fat of animal origin. Please supply in the text and References list.

Materials and Methods:

  1. Please divide text in 2.1 part into two parts: 2.1. Sampling, 2.2. Analysis of TFAs (or Determination ……………..). Also description of analysis is not sufficient. Please supply more information: instrument and column - detailed information should be provided, qualitative and quantitative analysis, standards, basic parameters of separation FAs.
  2. Position of ISO 15304-2007 was not included in References, please supply.

Results and Discussion:

  1. The results in part 3.1. are not properly described. Authors not presented which trans FAs were detected (C18:1 9 trans, or C18:1 10 trans). Which structural isomers were detected in fat extracted from burger (ruminant ?) and other products for example Fries.
  2. There is many studies (from different countries) about the presence of TFAs in pastries and confectionery products, please include and discuss.
  3. Authors not presented results of whole fatty acid profile in fat extracted from studied products (or calculated on 100 g products) why. As is underlined in many published papers there is a problem with increased consumption of saturated FAs/
  4. I do not understand the reason of presenting Table 1. Authors stated “the average BMI 228 values ranged from 22.4 to 23.4 for fast-food consumers and from 21.4 to 23.3 for non-229 consumers”. So what, there is no health problem related to the consumption. If the Authors would like to leave this table, please rethink and comment better.
  5. Figure 4 is OK and should be better commented in conclusion section and abstract.
  6. Maybe I am wrong (I am sorry) however I not found information about total intake of TFA by respondent (from whole fast food products which were consumed by them). It would be valuable information for readers.

Conclusions

  1. In first sentence Authors said “ This study investigated the daily intake of TFA through fast-food consumption 276 among the Yerevan population”. Please introduce intake of TFA per day or week or month …..
  2. Conclusions should be more compact. Please present the most essential and impressive sentences resulted from yours study.

Author Response

Comments and Suggestions for Authors

Trans-fatty acids in fast-food and intake assessment for Yerevan’s population, Armenia

Topic of manuscript is valuable, up- to date and probably would gather great audience. However I have doubts regarding preparation of manuscript and description of results. I found some shortcomings, weaknesses and doubts which should be revised or/and clarified during the review. My general and specific comments are given below:

General comment:

Firstly, it is serious mistake to present information about trans fatty acids arising in biohydrogenation PUFA in rumen (in ruminants) and trans fatty acids (FAs) formed during industrial hydrogenation of vegetable fats (or heating during frying process) in one sentence (Lines 31-34). Please revise and include correct information from literature. Naturally arising trans FAs (from milk fat or from beef /lamb meat) not cause health problems. For example vaccenic acid (C18:1 11trans) has beneficiary effect. The serious cardiovascular problems resulted from consumption of products containing industrially hydrogenated fats. You can find more information in other study.

Re: Authors are thankful for your important considerations and comments. Taking into consideration your comment, this part of the introduction was edited.

Detailed comments:

Comment: Introduction:

  1. Lines: 35-38: Please include also the latest research findings about the presence and content of trans FAs and their functional role in many products: https://doi.org/10.3390/nu13093087

Re: Following the Guide for the authors, the mentioned paper has not been included in the revised version of the manuscript, since it was irrelevant to this study.

  1. Line 66 it should be also included information about Commission Regulation (EU) 2019/649 of 24 April 2019 amending Annex III to Regulation (EC) No 1925/2006 of the European Parliament and of the Council as regards trans fat, other than trans fat naturally occurring in fat of animal origin. Please supply in the text and References list.

Re: Thank you for the very useful advice for the paper. Based on the comment, the reference was cited and added to the reference list.

Comment: Materials and Methods:

  1. Please divide text in 2.1 part into two parts: 2.1. Sampling, 2.2. Analysis of TFAs (or Determination ……………..). Also description of analysis is not sufficient. Please supply more information: instrument and column - detailed information should be provided, qualitative and quantitative analysis, standards, basic parameters of separation FAs.

Re: Thank you for pointing this. Taking into consideration the comment, the appropriate changes were done and the required details were included.

  1. Position of ISO 15304-2007 was not included in References, please supply.

Re: Thank you. The reference has been included in the list.

Comment: Results and Discussion:

  1. The results in part 3.1. are not properly described. Authors not presented which trans FAs were detected (C18:1 9 trans, or C18:1 10 trans). Which structural isomers were detected in fat extracted from burger (ruminant?) and other products for example Fries.

Re: Thank you for your comment. The main goal of this study was to assess the daily intake of TFAs via fast-food consumption. It was the first time in Armenia to carry out this kind of investigation. Hence, there was a need to have data not only on product consumption but also on the total content of TFAs expressed as grams per grams of the product. Moreover, it is worth to mention that exited regulatory requirements are set for the total content of TFAs and not for the specific isomers. Therefore, the authors would like to mention that in this manuscript the data on the total content of TFAs in fast foods are presented and discussed.   

  1. There is many studies (from different countries) about the presence of TFAs in pastries and confectionery products, please include and discuss.

Re: Thank you. Following to your comment, some studies in which the data on the total TFA were presented for 100 gram of the product (i.e. TFA g per 100 g of product) have been considered and added in the reference list. 

  1. Authors not presented results of whole fatty acid profile in fat extracted from studied products (or calculated on 100 g products) why. As is underlined in many published papers there is a problem with increased consumption of saturated FAs/

Re: Thank you for the comment. As it was mentioned in the previous comment’s response, to assess the daily intake of TFAs the data on TFAs content in the product were required. The TFAs analysis were done in the internationally accredited laboratory. Firstly, the TFA contents in fat extracted from studied products were determined, then expressed as “gram per 100 grams of product” based on the fat content of the product.

  1. I do not understand the reason of presenting Table 1. Authors stated “the average BMI 228 values ranged from 22.4 to 23.4 for fast-food consumers and from 21.4 to 23.3 for non-229 consumers”. So what, there is no health problem related to the consumption. If the Authors would like to leave this table, please rethink and comment better.

Re: Thank you for your comment. Based on the reviewers’ comments, the parts related to BMI were removed from the revised manuscript.

  1. Figure 4 is OK and should be better commented in conclusion section and abstract.

Re: Since the parts related to BMI were removed from the revised manuscript, this figure has been removed as well.

  1. Maybe I am wrong (I am sorry) however I not found information about total intake of TFA by respondent (from whole fast food products which were consumed by them). It would be valuable information for readers.

Re: This information was presented in 3.3 section as follows: “In the case of cumulative intake, the average DI of TFA is equal to 0.303 g/day…”.

Comment: Conclusions

  1. In first sentence Authors said “ This study investigated the daily intake of TFA through fast-food consumption 276 among the Yerevan population”. Please introduce intake of TFA per day or week or month …..

Re: All the data regarding the TFA intake are presented in the format of “per day” (or in the format of g/day).

  1. Conclusions should be more compact. Please present the most essential and impressive sentences resulted from yours study.

Re: Thank you. This section of the manuscript was edited.

Reviewer 2 Report

This paper is of interest to Armenia for monitoring purposes and adds to the global body of knowledge around the TFA content of foods, specifically fast food. However, the paper requires significant revision in the methods, results and discussion to improve the clarity of the paper. I would suggest the authors consider this paper becoming two papers as the section on BMI/obesity does not fit the aims of the paper. There is also no mention of ethical approval for the FFQ and 24-hour recall. My specific comments are in the attached report.

Peer review report

Trans-Fatty Acids in Fast-Food and Intake Assessment for Yerevan’s Population, Armenia

General concept comments

This study aimed to determine fast food consumption in the population and assess dietary exposure to TFA. A strength of this paper is that it is the first paper of its kind investigating TFA in Armenia and the topic is of interest following the release of the WHO action strategy to eliminate TFA from the global food supply. There are also new regulations restricting TFA for Armenia therefore this paper appears timely.

In general, there are a number of areas of weakness in this paper.

  1. The methods need considerably more clarification and detail about the sample and study FFQ and 24-hour recall. These are huge pieces of work, yet the methods cover this section very briefly. Did the FFQ and 24-hour call just investigate fast food consumption? Was the 24-hour recall just completed for one 24-hour period? More questions below.
  2. The results section contains information that is more ideally suited for the discussion.
  3. It is also unclear to me why the section on BMI/obesity is included when the paper aim is related to TFA. Also, the BMI data shown indicate BMI within the normal range. This could be addressed in a separate paper.

The references seem to be appropriate and recent.

Specific comments

Title - needs to reflect obesity? Or remove this from the paper

Appear a 'first of' type of study, therefore, it makes the study unique.

L18-19 Unclear about the inclusion of this sentence when this is not an aim of the study.

L77-79 This needs to be clearer. It is stated that the regulation has a limit of 20% then there is a new regulation of 2%. Is this after the original regulation? I think the TR CU abbreviation should be detailed and explained further – is this a national regulation or a food industry-controlled mandate? It seems to be older (2011) than the EUFA 2014 20% regulation. More clarification is needed.

L100 Do grocery stores sell fast foods? Please explain. How did you decide which fast foods to include? What is in the pie? Do some of the foods contain ruminant TFA e.g. from meat/butter?

L115 2.2 Heading needs to allude to data collection of what? e.g. Dietary intake data collection

L117 Need to explain more about 'the 24-hour recall' - when was this done? How was this done? Was this part of the current study or not? What was the sample size? What questionnaire did you use? L116-134 this section needs revision as it is not clear. How was the sample determined, how did you recruit the sample, more detail is needed on how the data was collected for the FFQ? and how this relates to the 24-hour recall.

L135-141 were the portion sizes for the FFQ or 24-hour recall? More clarity needed, suggest if both are part of the study they are described separately and more fully. Perhaps they should be two separate papers.

There is no description of the section related to BMI in the method.

L165 Under the Results heading - the first two sentences appear to have come from the author guidelines?? Need to delete.

Also, the results should only contain the current study results therefore the sentences that describe results from other papers need to be moved to the discussion. Why is there no discussion section?

3.2 Detail under this heading needs to be explained in the methods. I suggest you either delete the section on BMI or provide more explanation of it (it is not in the aims). The former is my recommendation.

Figure 2 Suggest the foods are listed in descending/ascending order to make it easier for the reader or you follow the order of foods in Figure 1.

Is it hot dog/burger or hot dog and burger - need consistency L206

L219 This should be in the discussion.

L274 Sentence needs removal

Author Response

Comments and Suggestions for Authors

This paper is of interest to Armenia for monitoring purposes and adds to the global body of knowledge around the TFA content of foods, specifically fast food. However, the paper requires significant revision in the methods, results and discussion to improve the clarity of the paper. I would suggest the authors consider this paper becoming two papers as the section on BMI/obesity does not fit the aims of the paper. There is also no mention of ethical approval for the FFQ and 24-hour recall. My specific comments are in the attached report.

Re: Authors express their thankfulness for your important and constructive comments. Following your comments, the section on BMI/obesity was removed from this manuscript.

The authors would like to mention that this study did not require ethical approval from an ethics committee, as it did not involve the collection of identifiers from the population who took part in the surveys. Carried surveys were anonymous.  

Peer review report

Trans-Fatty Acids in Fast-Food and Intake Assessment for Yerevan’s Population, Armenia

General concept comments

This study aimed to determine fast food consumption in the population and assess dietary exposure to TFA. A strength of this paper is that it is the first paper of its kind investigating TFA in Armenia and the topic is of interest following the release of the WHO action strategy to eliminate TFA from the global food supply. There are also new regulations restricting TFA for Armenia therefore this paper appears timely.

In general, there are a number of areas of weakness in this paper.

Comment:

  1. The methods need considerably more clarification and detail about the sample and study FFQ and 24-hour recall. These are huge pieces of work, yet the methods cover this section very briefly. Did the FFQ and 24-hour call just investigate fast food consumption? Was the 24-hour recall just completed for one 24-hour period? More questions below.

Re: Thank you for the comment. Indeed, these two questionnaires are huge pieces of work. They each represent a different study. The 24HR has been conducted in the framework of a Total Diet Study (TDS) to collect food consumption data among the adults residing in Yerevan city. Meanwhile, the FFQ was particularly designed to collect fast-food consumption data among the adult population of Yerevan. The relationship between these two questionnaires is that FFQ fast-food list was developed based on the most consumed fast foods cited in the 24HR. For better clarification, the description of 24HR will be removed from the paper and the FFQ will be described in more detail.

  1. The results section contains information that is more ideally suited for the discussion.

Re: Thank you for pointing this out. The section name was changed to “Results and Discussion”. It was a typo, as there isn’t a separate section for the discussion.

  1. It is also unclear to me why the section on BMI/obesity is included when the paper aim is related to TFA. Also, the BMI data shown indicate BMI within the normal range. This could be addressed in a separate paper.

The references seem to be appropriate and recent.

Re: Thank you. Following your comments, the section on BMI/obesity was removed from this manuscript.

Specific comments

Comment: Title - needs to reflect obesity? Or remove this from the paper

Appear a 'first of' type of study, therefore, it makes the study unique.

Re: Following your comments, the section on BMI/obesity was removed from this manuscript.

Comment: L18-19 Unclear about the inclusion of this sentence when this is not an aim of the study.

Re: The section on BMI/obesity was removed from this manuscript, so the sentence was removed too.

Comment: L77-79 This needs to be clearer. It is stated that the regulation has a limit of 20% then there is a new regulation of 2%. Is this after the original regulation? I think the TR CU abbreviation should be detailed and explained further – is this a national regulation or a food industry-controlled mandate? It seems to be older (2011) than the EUFA 2014 20% regulation. More clarification is needed.

Re: Thank you for the comment. The authors would like to mention that Armenia is a member of the Eurasian Economic Union (EAEU) (previously called Customs Union: CU). Therefore, Armenia follows all the requirements of the Technical Regulations of the Customs Union (TR CU) as national regulation.

For oils and fat, there is one technical regulation: TR/CU 024/2011, in which several changes were done during recent years. Particularly, according to the latest changes, the limit for “trans-isomers of fatty acids” has changed from 20% to 2% of the product’s total fat content.

Based on the aforementioned information, some changes were done in the mentioned part of the manuscript.   

Comment: L100 Do grocery stores sell fast foods? Please explain. How did you decide which fast foods to include? What is in the pie? Do some of the foods contain ruminant TFA e.g. from meat/butter?

Re: Thank you for the comment. Yes, grocery stores sell fast-foods too. The variety of the studied products include the main types of fast foods consumed by the studied population. The information from the previously conducted 24HR recall survey was taken into account. Besides, the scientific literature data on the main food sources of TFAs were considered as well. The total TFA content were determined in all the investigated products.    

Comment: L115 2.2 Heading needs to allude to data collection of what? e.g. Dietary intake data collection

Re: The heading name was changed to “Food consumption data collection”.

Comment: L117 Need to explain more about 'the 24-hour recall' - when was this done? How was this done? Was this part of the current study or not? What was the sample size? What questionnaire did you use? L116-134 this section needs revision as it is not clear. How was the sample determined, how did you recruit the sample, more detail is needed on how the data was collected for the FFQ? and how this relates to the 24-hour recall.

Re: Thank you for the comment. The 24HR was done in 2018-2019. The data collection period of 24HR lasted for one year, from October 2018 to September 2019. The 24HR was not part of this current study. It was a single 24 HR and was a separate study conducted in the framework of a Total Diet Study (TDS). The sample size was calculated to represent the whole population of Yerevan. It consisted of 1272 respondents, ranging from 18 to 65 years old. 24HR surveys were done by well-trained interviewers via face-to-face and telephone interviews. The questionnaire was designed to collect an estimate of the mean intake of foods and beverages consumed during the past 24 hours.

Comment: L135-141 were the portion sizes for the FFQ or 24-hour recall? More clarity needed, suggest if both are part of the study they are described separately and more fully. Perhaps they should be two separate papers.

Re: The mentioned portion sizes were for the FFQ surveys. The paragraph has been changed for better clarification.

Comment: There is no description of the section related to BMI in the method.

Re: Following your comments, the section on BMI/obesity was removed from this manuscript.

Comment: L165 Under the Results heading - the first two sentences appear to have come from the author guidelines?? Need to delete.

Also, the results should only contain the current study results therefore the sentences that describe results from other papers need to be moved to the discussion. Why is there no discussion section?

Re: Thank you for pointing this out. The section name was changed to “Results and Discussion”. It was a typo since there isn’t a separate section for the discussion.

Comment: 3.2 Detail under this heading needs to be explained in the methods. I suggest you either delete the section on BMI or provide more explanation of it (it is not in the aims). The former is my recommendation.

Re: Thank you for the important recommendation. Following your comments, the section on BMI/obesity was removed from this manuscript.

Comment: Figure 2 Suggest the foods are listed in descending/ascending order to make it easier for the reader or you follow the order of foods in Figure 1.

Re: Thank you for the suggestion. The appropriate changes were done.

Comment: Is it hot dog/burger or hot dog and burger - need consistency L206

Re: It is hot dog and burger. So, the appropriate change has been done.

Comment: L219 This should be in the discussion.

Re: Thank you for pointing this out. In the revised version of the manuscript, the section name is changed to “Results and Discussion”.

Comment: L274 Sentence needs removal

Re: Thank you. The sentence was removed.

Round 2

Reviewer 1 Report

I found that manuscript was substantially revised by Authors. I also appreciate answers on all comments also those where the authors do not agree with my opinion. Thank for scientific discussion.

I have only one suggestion which I introduced in my first comment in previous revision. Please consider to insert some information about beneficial role typical for dairy product vacenic acid C18:1 11trans (in Introduction part). In my opinion this scientific fact should not be ignored. I accept your answer about not presnting whole profile of FAs in the manuscript.

Author Response

The authors express their gratitude to the editor and reviewer for their time and valuable considerations. 

Reviewer 1

Comments and Suggestions for Authors

Trans-fatty acids in fast-food and intake assessment for Yerevan’s population, Armenia

I found that manuscript was substantially revised by Authors. I also appreciate answers on all comments also those where the authors do not agree with my opinion. Thank for scientific discussion.

I have only one suggestion which I introduced in my first comment in previous revision. Please consider to insert some information about beneficial role typical for dairy product vacenic acid C18:1 11trans (in Introduction part). In my opinion this scientific fact should not be ignored. I accept your answer about not presnting whole profile of FAs in the manuscript.

Re: Thank you for the very useful suggestion for the paper. Based on the comment, the required information was included in the introduction.